# Checkpoint Kinase 1 Is a Key Signal Transducer of DNA Damage in the Early Mammalian Cleavage Embryo

**DOI:** 10.3390/ijms24076778

**Published:** 2023-04-05

**Authors:** Vladimír Baran, Alexandra Mayer

**Affiliations:** 1Institute of Animal Physiology, Centre of Biosciences, Slovak Academy of Sciences, Šoltésovej 4, 040 00 Košice, Slovakia; 2Department of Obstetrics and Gynecology, First Faculty of Medicine, Charles University, 12000 Prague, Czech Republic; alexandra.mayer@vfn.cz

**Keywords:** cleaving embryo, Chk1 kinase, cell cycle checkpoint, DNA damage

## Abstract

After fertilization, remodeling of the oocyte and sperm genome is essential for the successful initiation of mitotic activity in the fertilized oocyte and subsequent proliferative activity of the early embryo. Despite the fact that the molecular mechanisms of cell cycle control in early mammalian embryos are in principle comparable to those in somatic cells, there are differences resulting from the specific nature of the gene totipotency of the blastomeres of early cleavage embryos. In this review, we focus on the Chk1 kinase as a key transduction factor in monitoring the integrity of DNA molecules during early embryogenesis.

## 1. Introduction

Restoring mitotic activity is a crucial objective for the early embryo in order to maintain the subsequent course of embryonic genome expression without any disruptions. This will guarantee the accurate transfer of genetic information to the next generation of the same species. The maintenance of gene integrity is important for all types of cells, but in the case of early embryonic cells, it is absolutely essential. Disorders of the cell cycle control mechanism in somatic cells result, in the optimal case, in the elimination of such cells or tissue parts. When the control mechanisms in pre-implantation embryos fail, this may greatly influence the overall course of embryogenesis and also lead to serious consequences for the eventual offspring [1]. In this regard, early embryos are likely to use a different “strategy” to cope with DNA damage compared to somatic cells [2]. Although the complete implications of DNA lesions created during reprogramming in early embryos are not well understood, it is clear that genomic stability must be maintained during the first initial stages, when the overall dynamic epigenetic modification of the genome occurs [3]. The maintenance of gene integrity is equally important with regard to the primary differentiation of totipotent blastomeres. The damage or formation of lesions in DNA molecules is not unusual even under physiological conditions. In addition, DNA damage in cycling cells can also be induced by so-called non-physiological factors of endogenous or exogenous origin. If DNA damage occurs in germ cells (oocytes or sperm) or the fertilized oocyte and the DNA lesions are not satisfactorily repaired, this can lead to the occurrence of chromosomal aberrations during early embryogenesis and eventually to genetic instability during subsequent embryonic development. Therefore, examining the events related to DNA damage response at the sub-cellular level, particularly in germline or embryonic cells, is of utmost importance [4].

## 2. Cell Cycle Checkpoints

Shortly after fertilization, a significant reorganization of sperm chromatin takes place, during which, maternal histones are substituted for protamines. DNA lesions are generated during paternal DNA demethylation and repaired during the first cell cycle after fertilization to prevent chromosome fragmentation, infertility or embryo loss [5]. Subcellular abnormalities that often occur after in vitro fertilization are associated with DNA damage, which is considered to be the main reason for the decrease in the success of embryonic development [6]. In connection with the epigenetic modification of the genome in the early stages of preimplantation embryos, DNA chain breaks occur [3]. This chromatin reorganization and the simultaneous initiation of the mitotic cell cycle after fertilization followed by relatively short cycles of blastomere cleavage suggest that early embryonic cells may exhibit specific responses to different forms of DNA damage. The preimplantation embryonic period is distinguished by a series of mitotic cleavages of blastomeres from the zygote to the blastocyst stage [7]. The first embryo cleavage takes a relatively long time compared to subsequent cleavage stages. Once the first mitotic cell cycle is complete, the embryo enters into a series of rapid cell cycles, leading to an increase in the number of blastomeres, but without significant cell growth [8]. Studies have shown that during early embryogenesis, preimplantation embryos exhibit higher levels of chromosomal abnormalities in the initial stages of cleavage compared to the late morula stage or blastocysts [9]. Thus, preimplantation embryos can acquire an aneuploidy phenotype already in early developmental stages, which points to the fact that these first mitotic cycles are more susceptible to chromosomal aberrations [10]. The monitoring of chromatin damage, the so-called cell cycle checkpoint, is therefore an essential aspect of the cell cycle [11,12], because DNA damage in early embryos can lead to an extension of the cell cycle delay, leading to a reduction in the cleavage rate during blastulation [13,14,15]. A good marker of DNA repair is the well-detectable phosphorylated form of histone H2A.X (designated as γH2A.X^S139^) and the enzyme PARP1 (Poly [ADP-ribose] polymerase1) [16,17,18].

Oocytes generally appear to be more resistant compared with sperm, probably due to the low oxygen concentration and high levels of antioxidants in the follicular fluid [19,20,21]. However, oocytes are naturally attacked by aging processes [22]. On the other hand, spermatozoa containing damaged DNA are able to fertilize fully matured oocytes, which leads to the logical assumption that oocytes take responsibility for the possible repair and remodeling of both the maternal and paternal genomes during the very early stages of embryogenesis [23,24]. This phenomenon can probably be explained by the fact that sperm are not transcriptionally active [25]. DNA damage inherited by any germ gamete must be repaired before the first S-phase after fertilization to reduce the risk of mutagenesis and the subsequent dysregulation of primary embryonic cell differentiation. This means that the embryo must “rely” on endogenous stocks of mRNA and protein transcripts accumulated during the growth phase of the oocyte up to the stage of expression of the overall embryonic genome. However, this timing is highly specific for individual animal categories. For example, it takes place at the two-cell stage in the mouse embryo and between the four- and eight-cell stages of the early morula in the human embryo (for review see [26]). In this context, the sensory proteins of damaged DNA in the fertilized oocyte are apparently of female origin, up to the stage of the activation of the embryonic genome [27]. It was confirmed that DNA damage transmitted by sperm can thus be recognized and repaired with the help of enzymes stored in the mature oocyte [28]. If there is any deficiency or inaccuracy in the repair process by the oocyte, it has the potential to create de novo mutations in the embryo, thereby fixing paternal DNA damage. This observation could provide evidence to support the idea that assisted conception procedures have the potential to increase the mutational load passed down to the offspring [29]. Although zygotes are able to recognize DNA damage, they have the potential to protect themselves from cell death through antiapoptotic protection, which may provide an opportunity for DNA repair and continued embryogenesis [30]. The preservation of the continued cleavage of an early embryo containing damaged DNA is enabled by a certain degree of tolerance of the G1/S and G2/M checkpoints in the zygote, or by a “specific” threshold for the level of this damage, until the creation of a fully functional apoptotic mechanism during the last stages of the preimplantation embryo [31]. On the other hand, such DNA lesions followed by correct repair may promote genome diversity in response to endogenous/exogenous causes. In the worst case, a high degree of genome instability in the initial embryonic cell cycles may lead to congenital disorders caused by chromosomal abnormalities [32]. Approximately half of blastocysts are estimated to contain genomic alterations that result in a high incidence of pregnancy losses [33]. Therefore, signaling molecules induced by damaged DNA in cleavage embryos lead to activation events that control the integrity of the genome.

Studies have shown that oocytes with a moderate degree of DNA damage can complete maturation, even though there is an increased number of lagging chromosomes in anaphase I. This can lead to a cellular phenotype of chromosomal fragments at the end of oocyte maturation in metaphase II [34]. It appears that oocytes may not be thoroughly successful in such repair in order to reach the metaphase II stage and become competent for fertilization and subsequently to form the maternal pronucleus. On the other hand, the completion of meiosis fails in oocytes with a high degree of DNA damage. Our recent study (focusing on the preimplantation development of mouse embryos after DNA damage induced before entry into the first S-phase) documented that even such a fertilized oocyte tolerates some degree of DNA damage, suggesting that the completion of the first cleavage stage is of utmost importance in ensuring the continuation of embryogenesis for as long as possible [35]. Several studies have addressed the cellular phenotypes of early embryos derived from DNA-damaged germ cells [36,37]. In this context, however, it will be necessary to add more detailed knowledge about the developmental consequences of these embryos, especially in the stages of primary differentiation of embryonic cells. Unlike the oocyte, the paternal contribution to the restoration of mitotic activity is limited to a highly differentiated, transcriptionally inert cell with minimal cytoplasmic content. It is evident that abnormalities in the structure of sperm chromatin, originating from spermatogenesis, can alter the chromatin configuration and result in DNA such as single-stranded or double-stranded DNA breaks [38]. After all, the resumption of mitotic activity and the initiation of embryonic genome expression take place in the maternally inherent environment of the fertilized oocyte. Nevertheless, cell cycle control checkpoints are limited in fully grown oocytes, which allows oocytes with DNA damage to resume meiosis unless the damage levels are severe [39,40]. Despite a certain degree of tolerance of maturing oocytes and very early embryos to DNA damage, the cell cycle signaling pathways are crucial for the activation of downstream effectors that control the integrity of embryonal genomes. From this aspect, early embryos overcome DNA damage using a different “strategy” compared with somatic cells [2]. In principle, the DNA damage response can result in three possible outcomes: (i) DNA damage repair; (ii) cell death mediated by the activation of the apoptotic pathway; and (iii) tolerance to the lesion, which can result in mutation or eventual carcinogenesis [41]. It was earlier documented that a few overexpressed embryonal genes are involved in DNA repair. In this sense, it seems that the repair of damaged DNA is the primary response (with certain tolerance to a low number of lesions) of the early embryo [42]. In the case of extensive or persistent DNA damage, the death of the embryo is the last resort to “protect” genomic integrity [14].

Cell cycle checkpoints play a key role in cell cycle regulation during early embryonic development, as they control the cycle sequence, genome integrity and the fidelity of major cell cycle events that determine the further course of mitotic division. This is especially important during the first cleavage stages of the early embryo, because these cell cycles are the longest during preimplantation development in mammals. The importance of this control over the course of the cycle is also confirmed by the fact that the embryonic genome is only fully activated after the S-phase of the one-cell embryo in mice and the four-to-eight-cell human embryo. Generally, the cell cycle control machinery consists of three major checkpoints that ensure the progression of the cell cycle. These include checkpoints G1/S, G2/M, and SAC (**s**pindle **a**ssembly **c**heckpoint). The so-called intra-S checkpoint can also be assigned to them—see Figure 1. The most sensitive checkpoint to DNA damage appears to be the G1/S checkpoint. Its activation prevents S-phase entry as well as DNA replication by the inhibition of Cdk2 or Cdk1 activation. The final control point of the cell cycle before entry into mitosis is the G2/M checkpoint. Chromosomal aberrations detected earlier can lead to the activation of this checkpoint and cell cycle arrest in the G2 stage. The SAC is the major control point in regulating the onset of cytokinesis. Its role is to prevent the premature separation of sister chromatids during metaphase–anaphase transition by delaying the anaphase onset [43]. Apparently, the most important focus of checkpoints is the control of the integrity of the DNA molecule as the central entity for the transfer of genetic information to the next generation of cells. If the DNA damage response affects cell proliferation, the cell cycle progression is reversibly inhibited to allow DNA repair. After successful DNA repair, the checkpoint is turned off and the cell cycle is restored [44]. However, precise replication of the genome during the S-phase is of fundamental importance, especially in one-cell embryos when the resumption of the mitosis cell cycle takes place. In this context, the double-stranded breakage of DNA is probably the most severe type of damage during this embryonic stage, as it can induce chromosomal instability and the failure of chromosomal remodeling [41]. If the genetic information is erroneously replicated during this process, it will lead to serious outcomes such as implantation failure, spontaneous abortion, genetic disease or embryo death [45]. It should be noted that developing human embryos are more sensitive to the consequences of DNA damage than early mouse embryos [46,47,48], which is likely related to evolutionary differences, with mouse embryos being more efficient at protecting their DNA integrity [49]. Initial experiments with exogenous DNA damage documented that the exposure of the oocyte, zygote or early embryo to γ-irradiation or laser microbeams [39,50] or certain chemical drugs, such as etoposide, bleomycin or neocarzinostatin [9,14,34,51], can cause damage that leads to delayed cleavage. As a result, cells that have been damaged may be able to complete the cell cycle, but they are more likely to experience an increase in micronuclei formation. When it comes to cleavage embryos, this damage can hinder development in the subsequent cleavage stage and frequently result in arrest prior to reaching the blastocyst stage. The application of UV irradiation or cisplatin (*cis-diammineplatinum(II)dichloride*) as DNA damage inducers updated the current understanding and knowledge in this field [14]. In this case, the treatment of two-cell embryos in the G2 phase caused DNA damage characterized by the increased phosphorylation of H2A.X histone. In addition, exposure to UV irradiation resulted in sustained G2/M arrest, whereas treatment with cisplatin enabled progression through mitosis and the subsequent activation of the G1/S checkpoint. Sperm-induced DNA damage caused a delay in DNA replication, leading to developmental retardation during progression into the two-cell embryonal stage. Furthermore, a significant portion of the embryos were arrested at the G2 to M phase transition [36].

Recent world statistics document rising rates of infertility [52,53]. DNA damage in sperm causes the fragmentation of the paternal chromosomes. Such an event leads to the random distribution of the chromosomal fragments over the two sister cells in the subsequent first cell division. In addition, DNA damage in sperm can lead to an unforeseen secondary effect of direct unequal cleavages, including the little-understood heterogoneic cell divisions. The consequence of these various types of damage is that embryos resulting from fertilization with damaged sperm often exhibit chaotic mosaicism. Such structural variations, aneuploidies and uniparental disomies induced by sperm DNA damage may compromise fertility, cause embryonic developmental delay and lead to rare congenital disorders. Sperm-induced DNA damage can cause a delay in DNA replication, resulting in retardation during the progression into the two-cell embryonic stage. Additionally, a significant portion of the resulting embryos may become arrested in G2/M-stage transition. The cause is the high proportion of aneuploidy of mitotic origin and subsequent disorders in chromosome segregation, resulting in 20 to 30 percent of blastocysts having the so-called mosaic phenotype [54,55,56]. Considering these facts and the still incomplete knowledge about the effectiveness or activity and control points of the cell cycle in maturing oocytes as well as in very early embryos, further research in this area is important, but it must be in connection with exogenous environmental factors that have the potential to damage DNA.

## 3. Chk1 as Regulator of DNA Damage Checkpoint

During the monitoring process of DNA integrity, numerous signal transduction events are coordinated during this process, with two key ones being the ATM-Chk2 (**A**taxia-**T**elangiectasia **M**utated kinase-**C**heckpoint **K**inase 2) and ATR-Chk1 (**A**taxia **T**elangiectasia and **R**ad3-related kinase-**C**heckpoint **K**inase 1) pathways – see Figure 2. The activation of these pathways is primarily critical for the appropriate coordination of cell cycle checkpoints and DNA repair processes [57]. In this context, the ATM and ATR kinases, as key mediators of the DNA damage response, have become part of an attractive therapeutic concept in cancer therapy in connection with the use of selective ATM and ATR inhibitors [58,59] that have the potential to be very effective against tumors with a high level of replication stress [60]. Chk1 is defined as a key downstream regulator of the ATR response and is phosphorylated by ATR on Ser-317 and Ser-345. Subsequently, activated Chk1 triggers the intra-S and G2/M-phase checkpoints [61]. It has already been documented that the protein kinase Chk1 is a main signal transducer of DNA damage checkpoints because it plays a key role in the control of the cell cycle. Evidence for this is that Chk1-deficient mice show abnormal cell cycle checkpoint function and early embryonic death [62]. In contrast to Chk1 ablation, mice with a Chk2 knockout are viable and appear normal, except for the fact that they display greater resistance to apoptosis [63]. These findings suggest that the ATR-Chk1 pathway is the only one absolutely required during early embryogenesis and that the activation of cell cycle checkpoints via Chk1—but not Chk2—is essential for development up to implantation. Subsequent results showed that Chk1 plays a role in various physiological regulatory processes, including apoptosis [64], cell cycle regulation in the process of fertilization [65] and oocyte postnatal maturation [66]. Mouse Chk1 protein levels are notably elevated during the zygote and two-cell embryo stages, indicating that Chk1 may be a maternal factor that plays a significant role in the fertilization process. In somatic cells, Chk1 is typically only activated in the presence of DNA damage or replication stress. In such conditions, one of its primary functions is to prevent the onset of mitosis by arresting the cell in the G2 phase until the damage has been repaired or replication has been completed [65]. This indicates that Chk1 gene mutations increase the activity of analogous mutant proteins even in the absence of genotoxic stress. Chk1 stability is controlled by its steady-state activity during unchallenged cell proliferation in order to maintain intrinsic checkpoints and ensure genome integrity and cell survival [67]. It has been recently shown that Chk1 (as well as Chk2) is expressed in mouse oocytes from the GV (germinal vesicle) to MII (metaphase II) stages and is localized subcellularly during oocyte maturation [68,69].

In mammals, the number of oocytes in females is limited. In this way, DNA damage repair can be responsible for the control of the oocyte pool and follicle formation in mammals. In this sense, it is very important to maintain the best quality of gametes up to fertilization. It is crucial for gametes to repair DNA damage, to avoid apoptosis. Such timely repair would prevent the transmission of genetic mutations to offspring [70]. This is important because DNA damage inherited from gametes or induced by the chromatin remodeling of pronuclei after fertilization thus has the potential for such repair in the zygote even before the first mitotic S-phase. This may prevent disruptions in the flow of blastomere cleavage and eliminate potential mutagenesis during later primary blastomere differentiation [26]. After the activation of zygotic transcription, the embryo becomes sensitive to DNA damage again, and it may use Chk1 to manage the DNA damage response, regulate cell cycle arrest and ensure genome stability [71]. During fetal development, oocytes self-induce hundreds of double-stranded DNA breaks (DSBs), which have to be repaired. However, oocytes are not very efficient at repairing DSBs. From this aspect, a great number of the oocytes are eliminated in a process that has been linked to the formation of follicles. Recent data suggest that the control of the oocyte pool and follicle formation is related to DNA damage repair monitored by Chk1 and Chk2 [72]. It is interesting that mild to moderate levels of DNA damage during meiosis do not significantly affect the completion of oocyte maturation [39,45,71,73]. The essence of this phenomenon has not yet been convincingly documented [4]. In embryonic oocytes, Chk1/Chk2 signaling to TRP53/TAp63 plays an important role in monitoring key meiotic events. These oocytes that reach the threshold level for unrepaired DNA breaks appear to be eliminated by a semi-redundant Chk1/Chk2 signaling pathway [74,75]. Chk1 has been shown to be activated by persistent DNA double-strand breaks in oocytes and to an increased extent when Chk2 is absent. Briefly, if Chk2 activity is absent, Chk1 is activated to an increased extent at a time when a higher incidence of DNA breaks persists [76].

The DNA damage response pathway is a network of cell cycle checkpoint signaling and DNA repair pathways that work in an integrated and coordinated manner to prevent the replication and transmission of high levels of endogenous and environmental DNA damage to the next generation of cells. The type of damage and the cell cycle phase where the damage occurs lead to the activation of different pathways. The activation of the DNA damage response pathway requires a coordinated effort between DNA repair pathways and cell cycle arrest signaling to facilitate repair and prevent the replication of damaged DNA during the G1 and S-phase checkpoints, as well as to prevent the transmission of damaged DNA to the next generation in the G2/M checkpoint. The control of the first cell cycle after fertilization significantly affects the resumption or correct transition to a regular mitotic process in the one-cell embryo. Interestingly, if the G2/M checkpoint is overcome by a Chk1 inhibitor in the case of maternally acquired proteins or by the delayed expression of paternal mutant Chk1, all subsequent somatic cell divisions have the potential to proceed normally [77]. It is possible that the G2 arrest mechanism of the fertilized zygote is hypersensitive to the action of Chk1 compared to later somatic cell divisions. Despite numerous experiments in this area, the question still remains as to whether the pre-implantation embryo has the potential to eliminate spontaneous DNA damage caused during the first mitosis to such an extent that it will not have a fundamental impact on the development of the post-implantation embryo or fetal organogenesis. The mechanisms controlling the onset of the cell cycle, and especially the initiation of mitosis, are complex and due to feedback loops [78]. An interesting consideration is whether this system in early embryos (at the time of embryonic genome activation) can adapt by modifying itself to a higher level of Chk1 sensitivity and activity in subsequent divisions. New studies suggest that the absence of Chk1 activity can lead to not only oxidative stress and apoptosis but also defects in spindle assembly and chromosome alignment. These findings highlight the crucial roles of Chk1 during the early stages of mouse embryo cleavage [79]. Although the roles of Chk1 have been reported in several models, its roles during early mouse embryonic development remain unknown. Since the phosphorylation activity of Chk1 is directed in various directions, all of the consequences of this activity are not yet known in detail. This is particularly accurate in the case of germ cells and early embryos, taking into account the different types and degrees of DNA damage. Different pathways in which checkpoint kinases are involved depend on the type of DNA damage and the cell cycle phase the damage occurs (for review see [2]).

The activation of the checkpoint during the S-phase is mainly triggered by a specific structure consisting of single-stranded DNA coated with replication protein A (RPA). This structure is recognized by ATR-interacting protein (ATRIP), which then recruits ATR to the damaged sites. Subsequently, ATR phosphorylates Chk1 to initiate the checkpoint response [80]. The embryonic S-phase is specific because it is relatively short compared to somatic cells. Therefore, it was hypothesized that this may render the cell more susceptible to replication stress, a condition in which replication fidelity and error repair are challenged. Even though mammalian embryos have a slower development rate than lower vertebrates, they still experience rapid DNA synthesis in the initial cleavages, leading to potential replication stress within the embryo [81]. In addition to the challenges of DNA replication, another crucial aspect of early embryogenesis is the precise regulation of the transcription machinery. This regulation must be closely coordinated with frequent DNA replication to mitigate the potential for DNA-damaging collisions between replication forks and RNA polymerases [82,83].

During cell proliferation, replication stress and DNA damage sensed by the protein kinases ATR and ATM result in a cascade of signaling events that can lead to a delay or arrest of the cell cycle. As with the G1/S checkpoint, the G2/M checkpoint is also triggered by the activation of ATM or ATR in response to DNA damage (double- or single-strand breaks). ATM and ATR activate Chk2 and Chk1, which subsequently inactivate Cdc25A phosphatase and phosphorylate Wee1 kinase. The ATR-Chk1-WEE1 pathway activates the control of both the intra-S and G2/M checkpoint control in response to replication stress and DNA damage, whereas the ATM-Chk2-P53 pathway preferentially controls the G1 checkpoint. Thus, activated Chk1/2 kinases inhibit Cdc24A, thereby arresting the cycling cell until the DNA damage is repaired [84,85,86]. In this way, the CyclinB-CDK1 kinase complex, which is responsible for mitosis-phase entry, remains inactivated, and the cell cycle is arrested at the end of the G2 phase until the lesion is repaired [84]. There is considerable and significant crosstalk between the two pathways, with Chk1 being a target of ATM and thus CDC25A being a target of Chk1, while both ATR and Chk1 can be targeted by p53 [75,87]. Replication stress refers to a process where replication forks slow down or stall temporarily due to various reasons such as the depletion of dNTPs or the presence of DNA lesions that interfere with DNA replication. In DNA-damaging events, ATR can be activated by various DNA-damaging factors, such as ultraviolet radiation, the depletion of dNTPs, topoisomerase poisons, alkylating agents, and DNA-crosslinking agents. Subsequently, Chk1 phosphorylates and inactivates both CDC25C and CDC25A, thus allowing the dephosphorylation and activation of the cyclin-dependent kinases CDK1 and CDK2, essential proteins in cell-cycle progression [88], preventing the removal of the inhibitory phosphorylation of CDK1and CDK2, respectively. The successful final progression of the cell cycle through the G1/S and G2/M phases ultimately depends on the activation of the CDK2/cyclin E and CDK1/cyclin A/B complexes, respectively. When WEE1, Chk1 or ATR is inhibited, CDK1 and CDK2 are activated so S-phase progression and mitotic entry occur without delay and without allowing DNA repair. In addition, ATR and Chk1 promote the repair proteins BRCA2 and RAD51 to be involved in the DNA repair machinery via recruitment to DSBs and stalled replication forks. In this process, Chk1 phosphorylates the key homologous recombination repair proteins [89]. In cycling cells with depleted *CHK1* gene, premature mitotic entry was observed when mildly under-replicated DNA occurred at the end of the S-phase [90,91]. This was due to a low threshold of origin firing [92]. In this way, unfinished replication can continue during the G2 phase and during G2/M transition, but with a lower intensity [48,93]. In somatic cells, ATR/Chk1 signaling is also associated with an exit from the S-phase and the expression of mitotic inducers, which prevents premature entry into mitosis under conditions of replication stress. However, insufficiently replicated DNA can persist in mitosis, leading to chromosomal instability. During unperturbed growth, the basal level of Chk1 activity is maintained during the S-phase. Under conditions of replication stress, Chk1 is activated at the end of DNA replication but is reactivated in the G2 phase, which may prevent mitotic entry. However, cells can overcome active Chk1 signaling and reach the onset of mitosis, revealing checkpoint adaptation. However, the continuation of cell division after Chk1 reactivation in G2 leads to arrest in G1 of the next cycle, thereby eliminating daughter cells from proliferation. In this context, it has also been documented that Chk1 reactivation during G2 relies on Cdk1/2 and a Plk1-dependent repair mechanism [94,95]. We have shown that early embryos, even with a lower level of newly synthesized DNA, can successfully proceed through the first cleavage process and continue into the higher stages of the preimplantation embryo. Thus, it seems that the efficiency of the DNA damage detection by the intra-S checkpoint in one-cell mouse embryo is limited. In this sense, the same situation was observed in the case of the G2/M checkpoint because the first cell cycle continued despite the incomplete DNA replication [35]. Analogous processes of control of the course of the cell cycle observed in somatic cells also take place in oocytes during reduction division or the maturation of the oocyte [96].

It has been repeatedly confirmed that, for the intra-S checkpoint, the Chk1 kinase is a key member controlling the mechanism of DNA synthesis in the cycling cell. Furthermore, it is evident that this kinase is also active at the G2/M checkpoint [97]. In the course of the cell cycle, the G2/M checkpoint is very important because it decides the entry of the cell into the dynamic phases of cell division. If this barrier is overcome (either by treatment with a Chk1 inhibitor in the case of maternal transmission or by the delayed expression of mutant paternal Chk1), all subsequent cell divisions can proceed. It appears that the arrest mechanism in the G2 phase of the zygote is specifically more sensitive to Chk1 activity compared to later cell divisions. The explanation may be the idea that this will prevent the transfer of damaged DNA or its immediate consequences to the daughter blastomeres or to the fetus. It is important to note that mutations in genes responsible for regulating genome stability can lead to cancer predisposition syndromes. However, Chk1 protein kinase, despite its crucial role in DNA damage signaling and checkpoint activation, has not been found to be affected by such mutations [77]. The Chk1 kinase, which plays an important role in this signaling, is considered an exception. Indeed, no germline mutations affecting Chk1 have been conclusively linked to human disease. Two new studies document that inherited mutations in the C-terminal domain of Chk1 are associated with fertilization disorders, even after in vitro fertilization (IVF). These mutations had the ability to induce zygotic arrest followed by pronuclear fusion failure, whereas the ectopic expression of wild-type Chk1 had no significant effect. In addition, it has been demonstrated that increased Chk1 activity caused by mutations arrests the G2/M transition of zygotes [98,99]. The question is whether blastomeres of the inner cell mass (ICM) and trophoblasts would retain their developmental potential. However, it is possible to state that treatment with a Chk1 inhibitor has the potential to pharmacologically address IVF failure in women suffering from infertility caused by inherited Chk1 mutations. Zhang et al. discovered dominant genetic mutations in *CHK1* that result in female infertility due to zygote arrest and the failure of pronuclear fusion. In addition, these mutations increase Chk1 activity, leading to G2/M arrest in zygotes [98].

The spindle assembly checkpoint (SAC) is an additional checkpoint that triggers metaphase arrest in the case of failure in the attachment of kinetochore–microtubules during mitosis [100]. The SAC is a key player in the mitosis of early embryonic cells. The deletion of SAC components (such as Mad2, Bub3 and BubR1) accelerates the metaphase–anaphase transition during the first cleavage in mouse embryos, leading to micronuclei formation, chromosome misalignment and aneuploidy, which result in reduced implantation and development delays [101]. Chk1 has been reported to have an important role in spindle assembly and chromosome alignment during mitosis [102,103,104]. This is evidenced by the fact that the inhibition or depletion of Chk1 induces premature mitosis with the appearance of fragmented chromosomes and aggravates the occurrence of aneuploidy [48,79,90,103,105,106]. Chk1 has been demonstrated to have critical functions in all established cell cycle checkpoints in oocytes. Its expression begins from the germinal vesicle stage and continues to the metaphase II stage, with localization in the cytoplasm and subsequent movement to the spindle after germinal vesicle breakdown from the pro-metaphase I (pro-MI) to MII stages in mouse oocytes. Chk1 depletion does not affect meiotic cell cycle progression after germinal vesicle breakdown and does not cause oocytes to be arrested in the MI stage with abnormal chromosome arrangement, but it decreases the expression of the spindle assembly checkpoint protein Mad2L1 (Mitotic Arrest Deficient 2-Like 1) and the coactivator of the anaphase-promoting complex/cyclosome Cdh1 (cadherin1). Its overexpression delays germinal vesicle breakdown. After germinal vesicle breakdown, oocytes progress through meiosis I, during which, the spindle assembly checkpoint is activated. This checkpoint prevents the separation of homologous chromosomes until all kinetochores are properly attached to spindle fibers, thus arresting oocytes in the pro-MI or metaphase I (MI) stages. These data suggest that Chk1 is involved in prophase I arrest and functions in G2/M checkpoint regulation in meiotic oocytes. Moreover, Chk1 overexpression can disrupt the regulation of the meiotic spindle assembly checkpoint and lead to errors in chromosome segregation [68]. Additional research has revealed Chk1’s involvement in the spindle assembly checkpoint and chromosome alignment through its regulation of kinetochore–microtubule attachment and the recruitment of BubR1 and Aurora B to kinetochores [79,106] as well as negative regulation of Plk1 by Chk1 [101]. More detailed studies have shown that Chk1 phosphorylates Mad2 at some sites, particularly S185 and T187 [107], and Chk1 regulates the subcellular localization and expression of Cdc20 and Mad2 required for the initiation of anaphase [108]. On the other hand, when Chk1 activity is decreased, several changes occur, including hyper-stable kinetochore–microtubules, the unstable binding of MCAK, Kif2b and Mps1 to centromeres or kinetochores, and the reduced phosphorylation of Hec1 by Aurora B [109].

Recent results indicate that the loss of Chk1 activity accelerates cell cycle progression at the first cleavage and thereafter disrupts the cleavage of early embryos into morulas/blastocysts. Chk1 has also been shown to be involved in the control of spindle assembly and chromosome alignment, probably through kinetochore–microtubule attachment regulation and the recruitment of BubR1 and Aurora B to kinetochores. This clearly highlights the very important role of Chk1 in the configuration and activity of mitotic spindles. In addition, the loss of Chk1 activity results in embryonic DNA damage and the worsening of oxidative stress. Unsolved DNA damage often leads to the induction of apoptosis and the autophagy of embryonal cells. These data indicate that the activity of Chk1 in early embryos participates in the DNA damage response and can affect the possibility of repairing DNA damage during embryonic cell cleavage. This confirms the importance of a dynamic balance in Chk1 activity during the early cleavage of the embryo [79]. According to recent studies, Chk1 and Chk2 are considered essential regulators of post-metaphase I events, including oocyte meiotic resumption. Studies have shown that inhibiting Chk1 does not have a significant effect on germinal vesicle breakdown (GVBD), but it does inhibit the first polar body (PB1) formation. However, it is interesting to note that the complete blocking or enhancement of PB1 extrusion could not be achieved through Chk1 inhibition. Inhibiting both Chk1 and Chk2 led to the impaired organization of the meiotic spindle and the condensation of chromosomes during both the MI and MII stages of oocyte development. These experiments in maturing oocytes showed that by inhibiting Chk1 and Chk2, γ-tubulin and securin localization is abnormal or absent, while P38 MAPK is activated. This confirms the importance of Chk1 in the MII stage of oocyte development. On the other hand, when Chk1/2 were inhibited, there was a decrease in the percentage of oocytes that were able to undergo second polar body extrusion and form pronuclei [110]. During early embryonic development, blastomeres with unresolved chromosome mismatches during the M-phase can escape the spindle assembly regulatory mechanism that controls kinetochore microtubule attachment and proceed to the cleavage of the embryo. This phenotype can lead to aneuploid daughter cells [111]. From our findings, it appears that early-stage embryos with mild DNA damage can successfully bypass the spindle assembly checkpoint (SAC) during the first cleavage stage, just as they do in subsequent stages up to the blastocyst stage [35].

## 4. Conclusions

The activity of the so-called checkpoints due to DNA damage during oocyte maturation is studied in more detail than in the case of the resumption of the mitotic activity of the early embryo. This logically follows from the fact that DNA damage in the oocyte primarily has potential for subsequent genomic instability that can manifest itself after the activation of the embryonic genome. There is a general consensus that the fully grown oocyte cannot launch a robust DNA damage checkpoint [4]. Despite the extensive research on early embryos, all the effects of gene instability on early embryonic development are still not fully explained. The question remains as to what contributes to the tissue- and gender-specific responses to DNA damage in blastocysts. In general, it must be based on the fact that embryonic development is manifested by specific cellular/molecular dynamics, which are different from the dynamics of somatic cells. These properties are more analogous to the characteristics of cancer cells in terms of high proliferation rates, increased replication stress and overall gene instability [2]. Considering gene mutations are primarily caused by the incomplete repair of DNA damage during early embryogenesis, the Chk1 and Chk2 kinases (as key signal transducers of DNA damage) could be included in the overall diagnostic (with the application of suitable biomarkers) or therapeutic concept (with a combination of radiotherapy and chemotherapy) in the prevention of congenital deformities or postnatal activation of neoplasias. Over recent years, the better understanding of DNA damage response pathways has contributed to the discovery of new treatment options in oncology. The role of a deficient DNA damage response in causing the genomic instability of cells and contributing to the development of cancer is increasingly evident [97,112].

## Figures and Tables

**Figure 1 ijms-24-06778-f001:**
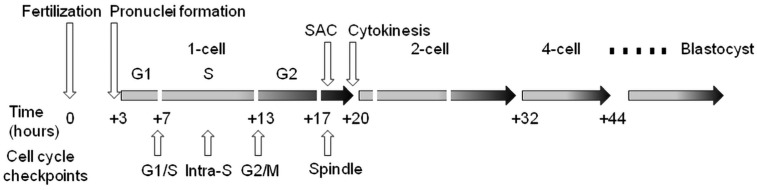
Cell cycle checkpoints in the temporal context of cleavage of the early mouse embryo. Time values are averages due to variability between individual lines of laboratory mice. SAC—spindle assembly checkpoint.

**Figure 2 ijms-24-06778-f002:**
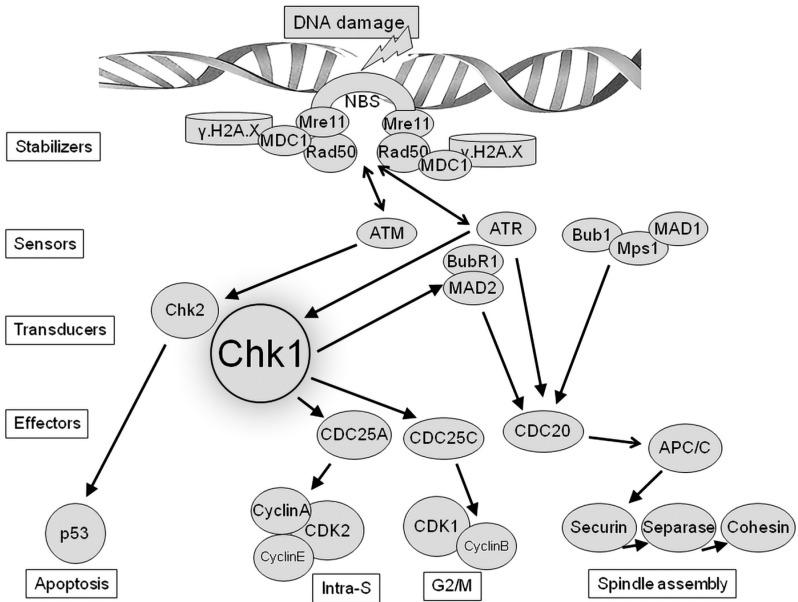
Simplified scheme of the Chk1 interactions in the DNA damage response.

## Data Availability

Not applicable.

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
