# Peer review of "Checkpoint Kinase 1 Is a Key Signal Transducer of DNA Damage in the Early Mammalian Cleavage Embryo"

_ijms, 2023, doi:10.3390/ijms24076778_

Round 1
Reviewer 1 Report
The manuscript deals with the problem of controlling the cell cycle during the first divisions in the early embryonic development of mammals. Unfortunately, the author could not clearly communicate his view on this problem, making the manuscript very difficult to read and understand.
Some concerns in detail:
- The manuscript needs to be better structured. For example, in the "Cell cycle checkpoints" section, the author began discussing the cell cycle checkpoints only in the fourth paragraph.
- The paragraphs in the manuscript are very long. This is because the paragraphs are concerned with too many ideas instead one particular idea per paragraph. For example, the first paragraph of the "Cell cycle checkpoints" section concerns three themes: the reorganization of sperm chromatin, dynamics of cleavage divisions, and chromosome aberrations in blastomeres. The second paragraph concerns two themes: the oocyte's role in the DDR of the early embryo and the specifics of DDR in the zygote.
- The author's argumentation is hard to follow. For example, the first paragraph of the "Cell cycle checkpoints" refers at first to sperm chromatin reorganization (Lines 42-45), then - to H2AX phosphorylation as a marker of DNA repair (Lines 50-51), and after that - again to "This chromatin reorganization" (Lines 52-53).
- Some phrases are too long and, thereby, incomprehensible. For example, "Although several studies have addressed the cellular phenotypes of early embryos derived from DNA-damaged germ cells [36, 37] even more detailed knowledge of the developmental consequences of preimplantation embryos in which DNA was damaged prior to repair is lacking of the first 123 mitotic activity." (Lines 120-124).
- The authors used the expression "In our previous study" on line 109 and " Our recent study" on line 116. This is inconsistent with the context of the review.
Reviewer 2 Report
In this review the author clearly describes the role of CHK1 kinase in zygote development, especially in DNA damage. The atuhor also talk about the DNA damage repair related proteins or complex in diffenent cell cycle period. It provides clear description and indication for further study of how DNA repair system maybe involved in the zygote or embryo development. This review is ready to be published.
Reviewer 3 Report
The review on Chk1 kinase DNA damage andin early cleavage mammalian embryo describes the mechanisms through which DNA damage may be repaired by cell cycle molecules. I read with interest the review and I think it deserves to be published. There are, however, several English language errors that make the text difficult to read. Thus I suggest the authors to submit an English revised version of the manuscript.
I describe some of my observations bellow:
Line 37 ….repaired, it event leads to infertility or chromosomal or ….it eventually ? Please reframe.
Line 70….”Oocytes generally appear to be more resistant compared with spermia (SPERM ?), probably due to the low oxygen concentration and high levels of antioxidants in the follicular fluid [19, 20, 21] and therefore they are protected for a long time until menopause”. Please explain this sentence. Considering that oocytes are stored in primordial follicles (no antrum, no follicular fluid-FF) from fetal life to menopause, how would they benefit from the FF environment, if they only encounter the FF when they are in an antral follicle? Please reframe.
Line 107 …effectors that effect the …… effect or affect? Please check.
Line 142 …dominant response to DNA damage in such embryogenesis DNA damage repair is preferred over cell division ………….this sentence is confusing….please reframe.
Line 161…The spindle assembly checkpoint (SAC)…do not need to write SAC in full again. Has been written in line 155.
Line 207….DNA integrity a multiple signal transduction events… Please check : DNA integrity multiple signal… (remove “a”)
Line 227…. , such a apoptosis [63], cell cycle regulation …such AS apoptosis
Line 241 …. with limited by the number of oocytes. Remove “by the”.
Line 253 … DSBs into their genome, please write “DSBs” in full, first time it appears in text.
Line 293….directions, the all concequences of this activity . the all concequences, please remove “the” and correct conSequences.
Line 299…. DNA coated with RPA protein. Please write RPA in full, first time it appears in the text.
Line 207 … has been shown to be that during development, WHEN the transcription ….. Please check the sentence.
Line 327… ecents, ...EVENTS? please check.
Line 332 … preventing that the removal of inhibitory. Please remove “that”.
Line 342… It was dues to a low …DUE , please remove “s”.
Line 430… It clearly implies that the role of Chk1 in spindle assembly configuration and activity. The sentence does not make sense. Remove “that”? Please check, reframe.
In Figure 1, please add the species to which the timing of events is shown.
Reviewer 4 Report
Review report
Chk1 kinase is key signal transducer of DNA damage in early cleavage mammalian embryo
By V. Baran
This manuscript gives an overview of pathways that in germ cells and early embryos act to minimize genomic DNA damage. The title suggests that it is focused on Chk1.
Major comments.
The manuscript should be improved for the English language. There are many mistakes, already in the title ‘Chk1 kinase is key signal transducer of DNA damage in early cleavage mammalian embryo’. This should be ‘Checkpoint 1 kinase 1 is a key signal transducer of DNA damage in the early mammalian cleavage embryo’, or ‘Chk1 is a key signal transducer of DNA damage in the mammalian cleavage embryo’.
In addition to grammatical errors there are numerous places in the manuscript where words are used two times while this is not necessary. For instance line 29 ‘… it is clear that early embryos must maintain genomic stability during the first initial stages of the preimplantation embryo…’ The part ‘of the preimplantation embryo’ is redundant and only leads to complexity of the sentence. The manuscript is littered with these. Another example line 21 ‘Control of the maintenance’ . Just ‘Maintenance’ suffices here.
Line 32 ‘totipotent blastomeres of the preimplantation embryo’. As totipotent blastomeres are by definition from a preimplantation embryo, ‘totipotent blastomeres’ suffices.
Line 115 ‘to form the female pronucleus a the zygote stage’. The part ‘at the zygote stage’ is redundant. These are just examples, there are more of such sentences in the manuscript. The readability would be greatly improved by changing these sentences.
Line 18 ‘The most important task of the early embryo’. This is very much debatable. What is the most important task, how is this evaluated’? ‘An important task’ is more suitable.
At various parts the reasoning by words such as ‘which points to the fact’ or ‘such a mechanism supports’ or ‘therefore’ do not make sense. For instance line 92 ‘such a mechanism support’. How does this mechanism support that assisted conception procedures enhance the occurrence of mutations? Apart from this ‘indicating’ together with ‘may’ is pleonastic. Another example line 105 “Therefore, the role of signaling pathways… genomic DNA’. The role of pathways cannot be crucial. This should be something like ‘Signaling molecules induced by DNA damage in cleave stage embryos leads to activation of events that control the integrity of the genome’
Line 193, reference 36 is rather old. More recent publications regarding sperm DNA damage and embryo development have been published and should be cited. For instance doi: 10.1186/s40659-022-00409-y; doi: 10.1126/sciadv.aaz7602; doi: 10.1111/andr.13277 ; doi: 10.1016/j.androl.2021.10.003
Ref 54 is a BioRxiv manuscript, so not peer-reviewed. My suggestion would be to remove this reference.
Figure 1 is a time line on mouse development, while to topic of the manuscript is human development. This figure should include a timeline human development, for instance comparing with the mouse time line and then in relative time periods. It is suggested to make the time periods to scale (so 1 hr a fixed distance).
The part line 248-266 describes that Chk1 can take over the function of Chk2 . This part should be directly after line 223 where it is described that that Chk2 KO mice are viable.
Lines 399-425 should be moved up, for instance to around line 221, where inactivation of Chk1 is discussed.
The final conclusion section ends (lines 470-479) with DNA damage response and treatment for cancer. Although interesting and important, one would expect a review manuscript on DNA damage in early embryos to end with ideas how to improve or control fertility.
Minor comments:
Line 28 important instead of importat.
Line 43 protamines are exchanged for histones, not nucleosomes.
Line 291 CHK1, while in the rest of the manuscript Chk1 is used.
The style of the references is not consistent. Eg references 2, 10, 73, 86 have different formats.